# Dialysis or a Plant-Based Diet in Advanced CKD in Pregnancy? A Case Report and Critical Appraisal of the Literature

**DOI:** 10.3390/jcm8010123

**Published:** 2019-01-20

**Authors:** Rossella Attini, Benedetta Montersino, Filomena Leone, Fosca Minelli, Federica Fassio, Maura Maria Rossetti, Loredana Colla, Bianca Masturzo, Antonella Barreca, Guido Menato, Giorgina Barbara Piccoli

**Affiliations:** 1Department of Obstetrics and Gynecology SC2U, Città della Salute e della Scienza-O.I.R.M., Sant’Anna Hospital, 10100 Turin, Italy; rossella.attini@gmail.com (R.A.); benedettamontersino@yahoo.it (B.M.); minellifosca@hotmail.it (F.M.); federica.fassio@hotmail.it (F.F.); bmasturzo@cittadellasalute.to.it (B.M.); guido.menato@unito.it (G.M.); 2Department of Surgery, Città della Salute e della Scienza-O.I.R.M., Sant’Anna Hospital, 10100 Turin, Italy; filomena.leone@unito.it; 3SCDU Nephrology, Città della Salute e della Scienza, University of Torino, 10100 Torino, Italy; mrossetti2@cittadellasalute.to.it (M.M.R.); lcolla@cittadellasalute.to.it (L.C.); 4Department of Medical Sciences, University of Torino, 10100 Torino, Italy; antonella.barreca@libero.it; 5Department of Biological and Clinical Sciences, University of Torino, 10100 Torino, Italy; 6Nephrology, Centre Hospitalier Le Mans, 72000 Le Mans, France

**Keywords:** chronic kidney disease, pregnancy, low-protein diets, vegetarian diets, small for gestational age baby, preterm delivery

## Abstract

Pregnancy is increasingly reported in chronic kidney disease (CKD), reflecting higher awareness, improvements in materno-foetal care, and a more flexible attitude towards “allowing” pregnancy in the advanced stages of CKD. Success is not devoid of problems and an important grey area regards the indications for starting dialysis (by urea level, clinical picture, and residual glomerular filtration rate) and for dietary management. The present case may highlight the role of plant-based diets in dietary management in pregnant CKD women, aimed at retarding dialysis needs. The case. A 28-year-old woman, affected by glomerulocystic disease and unilateral renal agenesis, in stage-4 CKD, was referred at the 6th week of amenorrhea: she weighed 40 kg (BMI 16.3), was normotensive, had no sign of oedema, her serum creatinine was 2.73 mg/dL, blood urea nitrogen (BUN) 35 mg/dL, and proteinuria 200 mg/24 h. She had been on a moderately protein-restricted diet (about 0.8 g/kg/real body weight, 0.6 per ideal body weight) since childhood. Low-dose acetylsalicylate was added, and a first attempt to switch to a protein-restricted supplemented plant-based diet was made and soon stopped, as she did not tolerate ketoacid and aminoacid supplementation. At 22 weeks of pregnancy, creatinine was increased (3.17 mg/dL, BUN 42 mg/dL), dietary management was re-discussed and a plant-based non-supplemented diet was started. The diet was associated with a rapid decrease in serum urea and creatinine; this favourable effect was maintained up to the 33rd gestational week when a new rise in urea and creatinine was observed, together with signs of cholestasis. After induction, at 33 weeks + 6 days, she delivered a healthy female baby, adequate for gestational age (39th centile). Urea levels decreased after delivery, but increased again when the mother resumed her usual mixed-protein diet. At the child’s most recent follow-up visit (age 4 months), development was normal, with normal weight and height (50th–75th centile). In summary, the present case confirms that a moderate protein-restricted diet can be prescribed in pregnancies in advanced CKD without negatively influencing foetal growth, supporting the importance of choosing a plant-based protein source, and suggests focusing on the diet’s effects on microcirculation to explain these favourable results.

## 1. Background

Pregnancy is increasingly reported in chronic kidney disease (CKD); this increase presumably reflects both higher awareness, in particular in the early CKD stages, previously often unrecognised, and an actual increase, in particular in later stages, due to a combination of improvements in materno-foetal care, significant changes in counselling, and a more flexible attitude towards “allowing” pregnancy in the advanced stages of CKD [1,2,3].

Success in the management of these high-risk pregnancies poses new clinical and ethical questions and highlights the existence of grey areas in which, due to the relative rarity of pregnancy in advanced CKD, clinical decisions often depend more on personal experience than on robust evidence [4,5,6,7].

One of the most crucial points in this context regards the indications for starting dialysis.

A growing body of evidence indicates that the outcomes of pregnancy are improved by long-hour intensive dialysis and suggests that “near normal” pre-dialysis blood urea nitrogen (BUN) is associated with a significantly longer duration of pregnancy, and with a decreased risk of intrauterine growth restriction [8,9,10,11,12].

These favourable results have occasionally been translated into an indication to start dialysis in the presence of a BUN level conventionally set as above 50 mg/dL. However, this indication is not fully supported by data in the literature, and while an emergency start of dialysis is intuitively associated with worse outcomes in pregnancy, in keeping with what is known about the overall population, no study comparing early or late dialysis start in pregnancy is yet available [13,14,15].

The choice of an early start of dialysis follows the logic of controlling uremic intoxication, in a moment in which attaining good metabolic balance is crucial [8,9,10,11,12]. However, this policy is in contrast with the indications for the non-pregnant population, which favour retarding dialysis start within a policy of “intent to defer” [16,17,18]. Comparisons are difficult to make: in contrast to pregnant women, the dialysis population is mainly elderly and has a high rate of comorbidity. The balance between the iatrogenicity of dialysis and uremic intoxication may be different in these two settings, thus underlining gaps in our knowledge.

Our group previously reported on the possibility of attaining stabilisation of the kidney function and proteinuria, and possibly favouring foetal growth in pregnant women with severe CKD or intense proteinuria, using a plant-based diet with moderate protein restriction, supplemented with a mixture of essential amino acids and their keto-analogues [19,20,21,22]. A similar case in Mexico has recently been reported [23]. One of the limits of our study was the impossibility of discriminating between the effect of the plant-based diet, the effect of protein restriction and the effect of ketoacid supplementation, in itself associated with a favourable anti-oxidative effect [24,25,26,27].

We would therefore like to report on a patient who was already on a moderately protein-restricted diet and who was put on a plant-based, non-supplemented diet, with the same protein intake. While outside of pregnancy the main goal of the diet is long-term (retarding progression, especially when the diet is started in early CKD stages), in pregnancy the priority is to stabilize the kidney function, avoiding the need to start dialysis.

## 2. Case

A 28-year-old woman, affected by stage-4 CKD, was referred to our obstetric-nephrology outpatient clinic in August 2016 at six weeks of amenorrhea. While previous caregivers had recommended the therapeutic interruption of pregnancy, on the account of the high risk of kidney function impairment and of preeclampsia, she was determined to continue her pregnancy.

At referral to our unit, this small, thin woman was apparently healthy. She weighed 40 kg (BMI 16.3), was normotensive, had no sign of oedema, had a creatinine serum level of 2.73 mg/dL, and creatinine clearance on 24-h urine collection of 28 mL/min. Her BUN was 35 mg/dL and proteinuria was 200 mg/24 h.

The first detection of kidney disease had occurred at one year of age, following investigations for poor growth, polydipsia and polyuria. Already on that occasion, kidney function was reduced to about half of the normal level, and ultrasounds showed agenesis of the right kidney and hypoplasia of the left one.

She had also undergone a kidney biopsy, which led to the final diagnosis of glomerulocystic disease (Figure 1), and had been on a regular nephrology follow-up since then. Kidney function showed a very slow decrease, with low-grade proteinuria and normotension. She had been on a moderately protein-restricted, mixed-protein diet since childhood, which she now self-managed, without recent dietary controls.

A treatment plan was made, including close monitoring of kidney function, blood pressure and proteinuria, psychological support, and switching from a conventional, mixed protein, moderately protein-restricted diet to the plant-based supplemented diet used in our centre [19,20,21]. In addition, she began taking a low-dose of acetylsalicylate to reduce the risk of preeclampsia and continued supplementation with 400 mcg of folic acid per day, which she had started at the time of the positive pregnancy test. At the following visit, the patient reported nausea, which she attributed to ketoacid and amino-acid supplementation, and said she had resumed her previous diet after a couple of days of trying the new one.

At 22 weeks of pregnancy, creatinine reached 3.17 mg/dL and BUN 42 mg/dL. Proteinuria had only minimally increased (from 200 to 400 mg/24 h). However since she was reaching a BUN level considered by some authors as a maximum threshold, starting dialysis was discussed again, and a plant-based diet option, this time without aminoacid and ketoacid supplementation, was proposed. Of note, the patient had chronic acidosis (HCO3 between 16 and 18 mmol/L without bicarbonate supplementation) and bicarbonate supplementation was started during pregnancy, stabilizing blood bicarbonate levels between 20 and 24 mmol/L.

In spite of the stable protein intake, the switch from a mixed-protein to a plant-based diet was associated with a rapid decrease in serum urea and creatinine; this favourable effect was maintained up to the 33rd gestational week, when a new rise in urea and creatinine was observed, along with a mild increase in proteinuria (Table 1, Figure 2). At 31 weeks of pregnancy the patient developed cholestasis and itching: transaminase levels were normal, while blood levels of bile acids were slightly increased (10.7 µmol/L). Ursodeoxycholic acid (450 mg twice/day) was added, but the following week bile acids reached 29.4 µmol/L.

Foetal growth was normal throughout pregnancy (Figure 3), and utero-placental Doppler flows were normal both at the umbilical and uterine arteries.

At 33 weeks + 5 days, considering the gestational age achieved, the progressive worsening of the renal picture and the patient’s desire to preserve her renal function, labour was successfully induced with the placement of a cervical ripening balloon and subsequent amniorexis and oxytocin infusion. A healthy female baby, adequate for gestational age, was born on the following day (33 weeks + 6 days, birth weight 1900 g, 39th centile according to the Italian reference charts). The Apgar score was 8 at 1 and 5 min and the baby, initially under observation in the neonatal intensive care unit on account of her prematurity, was discharged in good health 10 days later.

At the mother’s final follow-up visit, two months after delivery, she was in good clinical condition, normotensive and with low-grade proteinuria. It is significant that as soon as she resumed her usual diet, her urea levels rapidly increased, even though her overall serum creatinine remained stable (2.82 mg/dL). At the child’s most recent follow-up visit (age 4 months), routinely scheduled for pre-term babies, development was normal, with a normal weight (50th–75th centile), thus making it unnecessary to schedule additional follow-ups, according to the routine policy of the hospital.

### 2.1. Diet

#### 2.1.1. Dietary Habits before Pregnancy

The patient reported that she had followed a moderately protein-restricted diet since childhood, mainly based on replacing the carbohydrates most often used in Italian cooking (pasta, bread, biscuits, etc.) with protein-free food. Her pre-pregnancy energy intake had been about 2000 kcal/day, with a daily intake of 0.8–1 g of protein/actual body weight, about 0.6–0.8 g of protein per ideal body weight, with wide day-to-day variations (0.6–1 g per ideal body weight). She said she often consumed high-calorie, nutritionally poor foods, such as chips, popcorn, sugary drinks, ice-cream and sweets, but did not drink alcoholic beverages.

#### 2.1.2. Plant-Based Diets

The initial prescription consisted in a plant-based supplemented diet (6 Ketosteril tablets per day), in keeping with the usual schemes reported in detail elsewhere [19,20]. Protein and energy prescriptions were not changed (energy intake: 2000 kcal/day; protein intake 0.8 g/ideal weight/day). As previously reported, Ketosteril was not tolerated as it caused nausea, and the patient therefore resumed the mixed-protein diet she had previously followed, until, due to an increase in urea, a second plant-based non-supplemented diet was prescribed, once more with the same energy and protein intake.

This latter diet included small quantities of milk in the morning, and an occasional yogurt as a snack during the day, with muesli, cereal or biscuits, and a small portion of animal-derived food three to four times per week at lunchtime. Less than 20% of the protein intake was of animal origin, thus reducing the overall acid load. The fibre intake was high, but it was not specifically evaluated, due also to the indication to vary the type of food as much as possible within the described indications.

Each main meal included cereal products (pasta, rice or other grains), beans, vegetables and fresh fruit, while dried fruit and nuts were eaten as snacks during the day.

In the absence of reference data on the reference urinary urea in pregnancy, the protein intake was regularly assessed by the dietician, who controlled the patient at least monthly.

## 3. Discussion

When dialysis should begin is a hotly debated question in the current management of advanced CKD in pregnancy.

While on the one hand the evidence now accumulating suggests that maintaining a near-normal pre-dialysis urea level is associated with pregnancy outcomes never previously possible in dialysis patients, on the other hand, the lack of advantage associated with early dialysis outside of the context of pregnancy has led to reconsider over-aggressive dialysis policies, suggesting that a “healthy” dialysis start may be associated with increased morbidity [8,9,10,16,17,18].

It is in this context that our limited experience with the dietary management of CKD pregnancy is discussed [19,20,21,22,23].

A dietary plan including adequate energy intake, balanced macronutrient and micronutrient distribution and different degrees of protein restriction is a useful tool for controlling the signs and symptoms of uraemia in patients with advanced CKD [28,29,30,31,32]. This strategy can serve to reduce the progression of CKD, and postpone the need for renal replacement therapy by correcting, at least in part, some of the metabolic derangements of CKD, and in particular acidosis and hyperphosphatemia, with a favourable impact on the clinical and social burden of end-stage renal disease (ESRD) [28,29,30,31,32]. In pregnancy the priority shifts from these medium-long-term goals to the short-term goal of avoiding the need for starting dialysis during gestation, with the hope that stabilizing kidney function is associated with a better materno-foetal outcome.

The experience in CKD pregnancy is limited: pregnancy in advanced CKD is still a rare event. Limiting protein is clearly in contrast with the widespread practice of increasing the mother’s protein intake over the physiological range, with the hypothesis that this may favour foetal growth. This is a policy which it now appears may even be deleterious, outside the CKD context; in fact, increasing protein intake over the indicated levels has been associated with retarded foetal growth and hypertensive disorders of pregnancy [33,34,35]. In such a context, there is usually a high intake of animal-derived proteins, and it is not possible to distinguish between the effect of the specific type of proteins and that of the additives and preservatives that are usually added to such aliments.

Furthermore, although the results obtained using plant-based diets in pregnancy are promising, the available evidence mainly comes from the same group, and regards moderately restricted, supplemented, plant-based diets. As a consequence, it is not possible to distinguish between the effect of each element (protein intake, supplementation with essential ketoacids and amino acids and quality of proteins).

In this context, we hope that the present case report will contribute to this discussion, by suggesting that a crucial item is in fact the quality of the proteins the patient eats. In the context of serum urea approaching a level that is considered by some experts as an indication for dialysis start, the plant-based diet led to a rapid reduction to a “safe level” with regard to the urea-based indication for dialysis start.

Our patient’s history is interesting: she has a congenital, slowly progressive kidney disease (glomerulocystic dysplasia), which occurred in the absence of the three main elements that are associated with a higher risk of adverse pregnancy outcomes, at each level of kidney functional impairment (hypertension, proteinuria and immunologic disease) (Figure 1). Therefore, her history can be taken as a good example of the nature of the challenges patients with advanced, non-immunologically-mediated CKD face in pregnancy.

Since childhood, the patient had been on a moderately protein-restricted diet, which included mixed proteins (in keeping with the long-standing indication that animal-derived protein should constitute at least half of total protein intake), with the support of protein-free food, which in Italy is furnished free to all CKD patients [36,37]. Therefore, the switch to a plant-based diet was made without changing her protein intake, and the biological modifications observed shortly after switching to a plant-based diet and shortly after resuming her usual dietary habits can be considered a good test of the effect of the quality of proteins (plant-based versus animal-derived) on stable protein intake.

Furthermore, since the patient did not tolerate the aminoacid and ketoacid supplements, the effect of the diet was not enhanced by the potential benefit of supplementation [24,25,26,27].

The graph in Figure 2 speaks for itself: urea levels dropped immediately after changing the main protein source and remained substantially lower up to the last phase of pregnancy, when a rapid increase in urea and creatinine levels occurred, along with a minor increase in proteinuria and some initial signs of cholestasis, as well as worsening of anaemia, in the absence of any sign of bleeding or inflammation.

The pattern is not classic superimposed preeclampsia (PE), or “maternal preeclampsia”, a term often employed to define the full blown picture of hypertension and proteinuria developing or worsening on the basis of a pre-existing maternal disease (very often CKD). However, this pattern suggests a non-specific CKD-induced placental-dysfunction, quite common in CKD, where proteinuria often increases or develops in pregnancy, with or without hypertension, and is associated with an increased risk of preterm delivery and intrauterine growth restriction [6,7,38,39,40,41,42]. The presence of a worsening of kidney function led to the decision to induce delivery, also due to the search for a compromise between maternal and foetal health.

Supporting the interpretation of a transitory pregnancy-induced event, urea levels rapidly dropped post-partum, and remained low until the previous diet was resumed, following the patient’s preferences (Figure 2).

Foetal growth was regular throughout pregnancy, and the baby was born with an adequate weight, adjusted for gestational age (39th centile), a very good result given also the mother’s small body size. The baby’s subsequent development was regular and her clinical history uneventful.

This isolated case is in line with previous reports by our group that suggested that the use of plant-based, supplemented diets is associated with better foetal growth than an unrestricted diet in patients with advanced CKD or intense proteinuria [19,20,21]. The reason for this advantage is not known. Plant-based diets are rich in anti-oxidants, lead to a lower acid load, and are probably able to induce a favourable selection of the intestinal microbiota. In addition, ketoacids have a specific anti-oxidant effect and may contribute to a lower urea level through their transformation into amino acids [24,25,26,27,43,44,45,46,47,48,49]. Plant-based diets, when nutritional deficits are avoided, are presently considered to be safe in pregnancy and their protein content is usually lower than that of omnivorous diets. While the field is only partly explored, overall the advantages seem to outnumber the shortcomings [50,51,52,53,54,55,56,57,58,59].

Once more, this case can help us distinguish between different effects, as it suggests that the reduction in the patient’s urea levels was closely connected to the quality of protein she consumed.

Interestingly, creatinine swings coincided with urea changes (Figure 2, Table 1); in fact, in a metabolically stable CKD patient, only urea is strictly linked to protein intake and is held to be influenced by diet, while serum creatinine is a more reliable marker of muscle mass and is considered to be less affected by diet. The changes in serum creatinine in fact suggest that a plant-based diet has a favourable effect on the kidney function, possibly via a vasodilator effect on microcirculation [19,20].

In summary, the present case is an addition to the scarce current literature on the dietary management of CKD pregnancies. Firstly, an analysis of what occurred supports the view that the quality is more important than the quantity of protein consumed in controlling the high urea levels commonly associated with stage 4–5 CKD. Secondly, it confirms that a moderate protein-restricted diet can be prescribed in pregnancies in advanced CKD without negatively influencing foetal growth. Thirdly, it suggests that focusing on the diet’s effects on microcirculation would allow us to better understand the underlying mechanisms of action of plant-based proteins.

Should these findings be confirmed in a larger series, they would point to a wider use of plant-based diets in pregnancy as a tool for stabilising kidney function and avoiding the need for starting dialysis, an important issue, particularly in settings in which access to this life-saving treatment is still unavailable or restricted.

## Figures and Tables

**Figure 1 jcm-08-00123-f001:**
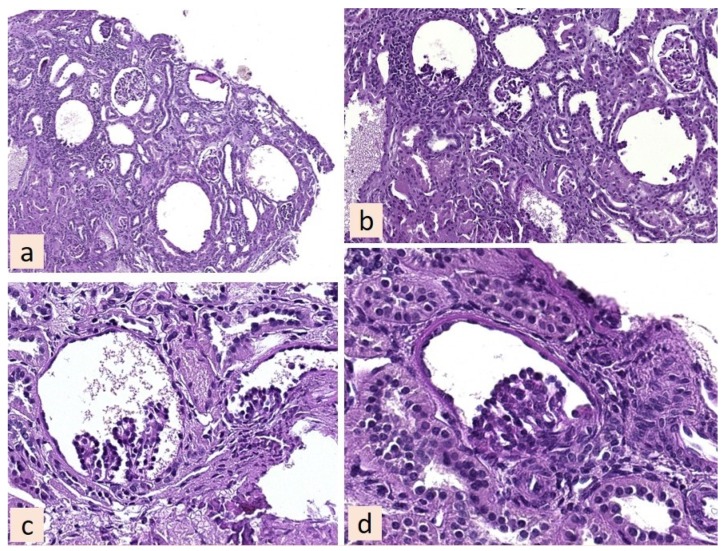
(**a**) Histological section of kidney shows glomerulocystic dysplasia, cyst size inversely correlates with the cellular component of the glomeruli until giant glomerulocysts with glomerulus remnants and complete cystic replacement of the tuft (Periodic acid-Schiff or PAS original magnification 100×). (**b**) Round glomerular cysts (PAS original magnification 200×). (**c**,**d**) Higher magnification showing rudimentary capillary tuft and dilatated Bowman’s space (PAS original magnification 400×). At her first visit to the kidney and pregnancy unit, the patient said that she was terrified by the idea of starting dialysis, but at the same time she did not want to terminate her pregnancy.

**Figure 2 jcm-08-00123-f002:**
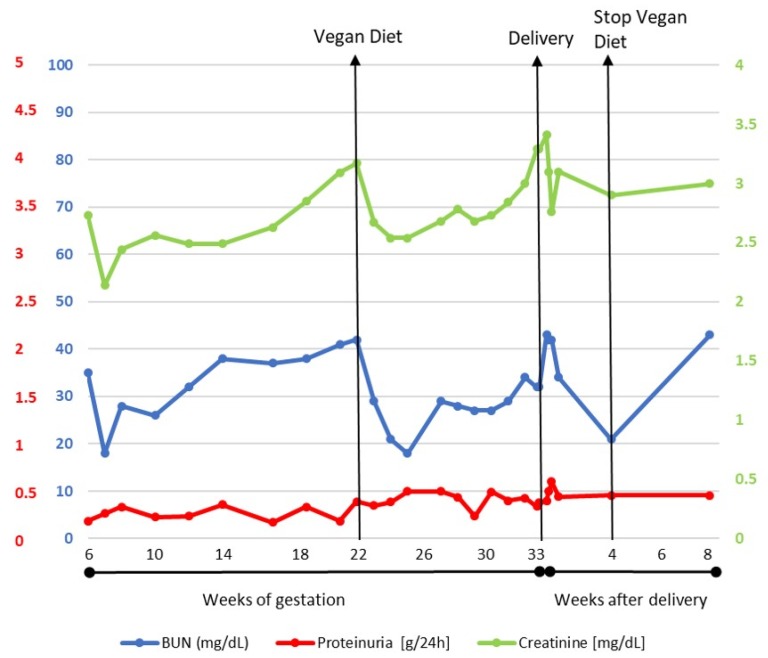
Trends of serum creatinine, BUN and proteinuria during and after pregnancy. BUN: blood urea nitrogen.

**Figure 3 jcm-08-00123-f003:**
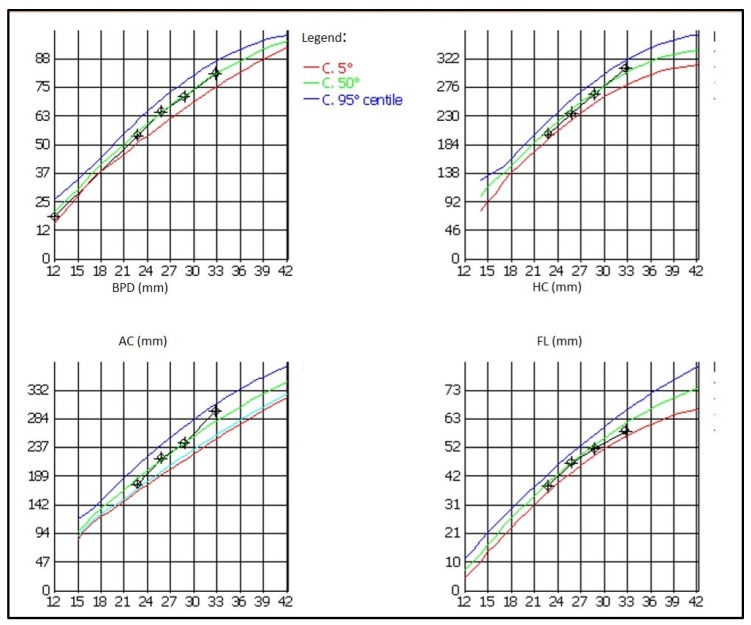
Foetal growth during pregnancy. BPD: biparietal diameter; HC: head circumference; AC: abdominal circumference; FL: femur length.

**Table 1 jcm-08-00123-t001:** Biochemical data from referral to delivery and after pregnancy.

	6 Weeks	12 Weeks	22 Weeks	30 Weeks	33 Weeks (Last Check-Up before Delivery)	3 Months after Delivery
sCr (mg/dL)	2.73	2.49	3.17	2.70	3.29	2.82
eGFR CKD-EPI (mL/min)	23	26	19	23	18	20
BUN (mg/dL)	35	38	42	27	32	45
Proteinuria (g/day)	0.200	0.385	0.419	0.532	0.364	0.500
Haemoglobin (g/dL)	13.9	12.7	11.2	11.0	9.1	10.4
Serum Albumin (g/dL)	4.3	4.5	4.3	4.1	3.3	3.7
Total Protein (g/dL)	7.0	7.1	6.8	6.9	6.1	NA
Calcium (mmol/L)	2.72	2.44	2.83	2.44	2.40	2.46
Phosphate (mmol/l)	0.98	0.91	1.14	0.97	NA	0.94
PTH (pg/mL)	NA	110	21	NA	NA	120
Vitamin D (ng/mL)	24.6	NA	54.6	NA	37	44.9
Vitamin B12 (pg/mL)	>2000	631	NA	NA	371	748
Folic acid (ng/mL)	>20.0	>20.0	NA	NA	>20.0	>20.0
Weight (kg)	40	42	44	45	46	41
Blood pressure (mmHg)	110/80	100/70	100/60	90/60	90/55	100/70
Therapy	Alphacalcidol 1 µg/day; ASA 100 mg/day; Folic acid 400 µg/day; Cobalamin 1000 IU/week	Alphacalcidol 1 µg/day; ASA 100 mg/day; Folic acid 400 µg/day;	Alphacalcidol 1 µg/day; ASA 100 mg/day; Folic acid 400 µg/day; HCO3 10 g/day	Cholecalciferol 50,000 IU/week; ASA 100 mg/day; Folic acid 400 µg/day; HCO3 10 g/day	Cholecalciferol 25,000 IU/week; Folic acid 400 µg/day; HCO3 10 g/day Ursodeoxycholic acid 900 mg/day.	Oral iron

Legend: BUN: blood urea nitrogen; eGFR: estimated glomerular filtration rate; PTH parathyroid hormone; ASA: acetyl salicylic acid.

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
