# Peer review of "Dialysis or a Plant-Based Diet in Advanced CKD in Pregnancy? A Case Report and Critical Appraisal of the Literature"

_jcm, 2019, doi:10.3390/jcm8010123_

Round 1
Reviewer 1 Report
The manuscript entitled "Dialysis or a plant -based diet…" is an original and interesting case report to manage pregnant CKD patients. The focus on plant-based diet is nowadays a emerging topic.
The manuscript is well written, clear and with excellent discussion, however ther are some issues that could be clarify.
Abstract : properly written
Background: Line 93 ; I suggest modify the sentence "avoiding need of dialysis" by delaying the CKD progression.
Case report description: Well expressed, except in Table I where some mistakes are apparent as Ca,& P, values are not real in mg/dL. Cr and creatinine Clearance could be delete, maintained CKD-EPI.
The diets : some suggestions are required explaining the reasons why dietary assessment record has not performed. This is a survey method to evaluate how the patient is adherent to the diet and also account the intake ratio between animal/vegetarian proteins are realize. An importan issue is the acid load in CKD patients.
Also to evaluate the fibber intake.
Discussion : Generally well performed with excellent guiding theme. Some relevant issues require some words to be included. Plant-based diets not only reduce BUN if not have important influence to correct the metabolic disarrangements in CKD as metabolic acidosis, improvement appetite, reduced P intake and finally reduce the progression of CKD in patients with GFR<30 ml/min.
By other hand, culinary techniques uses in plants and vegetables, play a great role to reduce K & P intake conducting to an improvement in cardiovascular events and CKD-MB management.
So in conclusion the manuscript has scientific value but the suggestion could improve the quality.
Author Response
Answer to reviewer 1:
Thanks for having taken the time to review our paper and fro the suggestions to improve it.
Please find in the following lines our answers, highlighted in red.
The manuscript entitled "Dialysis or a plant -based diet…" is an original and interesting case report to manage pregnant CKD patients. The focus on plant-based diet is nowadays a emerging topic.The manuscript is well written, clear and with excellent discussion, however there are some issues that could be clarify. Abstract : properly written
Thanks for your kind appreciation.
Background: Line 93 ; I suggest modify the sentence "avoiding need of dialysis" by delaying the CKD progression.
Indeed, delaying the CKD progression is a goal in the medium long term, while during pregnancy we just intend to avoid the need of dialysis; this was better clarified in the paper, as follows.
While outside of pregnancy the main goal of the diet is in the long term (retarding progression, especially when the diet is started in early CKD stages), in pregnancy the priority is to stabilize the kidney function, avoiding the need to start dialysis; indeed.
Case report description: Well expressed, except in Table I where some mistakes are apparent as Ca,& P, values are not real in mg/dL. Cr and creatinine Clearance could be delete, maintained CKD-EPI.
Thanks: we corrected the units, and we omitted Cockroft and Gault formula; however, due to the lack of identification of a gold standard formula in pregnancy, we also left the information on serum creatinine.
The diets : some suggestions are required explaining the reasons why dietary assessment record has not performed. This is a survey method to evaluate how the patient is adherent to the diet and also account the intake ratio between animal/vegetarian proteins are realize. An important issue is the acid load in CKD patients. Also to evaluate the fibber intake.
The dietary intake was regularly assessed by the dietician; since the diet was plant based, about 90% of the protein intake was of vegetable origin; this was clarified in the paper; the acid load was not assessed, but the fiber intake was overall high. This information was added in the paper.
This latter diet included small quantities of milk in the morning, and an occasional yogurt as a snack during the day, with muesli, cereal or biscuits, and a small portion of animal-derived food 3-4 times per week at lunchtime. Less than 20% of the protein intake is of animal origin, thus reducing the overall acid load. The fibber intake is high, but it was not specifically evaluated, due also to the indication to vary as much as possible the type of food, within the described indications.
Each main meal included cereal products (pasta, rice or other grains), beans, vegetables and fresh fruit, while dried fruit and nuts were eaten as snacks during the day.
In the absence of reference data on reference urinary urea in pregnancy, the protein intake was regularly assessed by the dietician, who controlled the patient at least monthly.
Discussion : Generally well performed with excellent guiding theme. Some relevant issues require some words to be included. Plant-based diets not only reduce BUN if not have important influence to correct the metabolic disarrangements in CKD as metabolic acidosis, improvement appetite, reduced P intake and finally reduce the progression of CKD in patients with GFR<30 ml/min.
By other hand, culinary techniques uses in plants and vegetables, play a great role to reduce K & P intake conducting to an improvement in cardiovascular events and CKD-MB management.
We fully agree with the reviewer on the general interest for diets in CKD, and we added some precisions in this issue; however, the goals in pregnancy may be different, and we clarified it as follows:
This strategy can serve to reduce the progression of CKD, and postpone the need for renal replacement therapy, by correcting at least in part some of the metabolic derangements of CKD, and in particular acidosis and hyperphosphatemia, with a favourable impact on the clinical and social burden of end-stage renal disease (ESRD) (28-32). In pregnancy the priority shifts from these medium-long-term goals to the short term one of avoiding the need for starting dialysis during gestation, with the hope that stabilizing kidney function is associated with a better materno-foetal outcome. Thanks again for your kind and keen remarks.
Reviewer 2 Report
On page 8 , under "Plant based diets" line 171 - authors mention "small portion of animal-derived food 3-4 times per week at lunch time". What animal derived food was used? How much of the 2000Kcal/day calorie intake and 0.8g/ideal weight/day of protein intake per day did the animal derived food account for?
On page 9, line 225 and 226, the authors mention that the patient may have had "maternal pre-eclampsia" that led to the decision to induce delivery.
Maternal Pre-eclampsia is a pregnancy complication characterized by high blood pressure and signs of damage to other organs such as liver and kidney. While the patient developed cholestasis and elevated creatinine, her blood pressure was only 90/55 mm hg suggesting against pre-eclampsia.
What other differential diagnoses were considered for evaluation of AKI? The patient's hemoglobin dropped by almost 2 units from 30 weeks to 33 weeks(although pregnancy can cause dilutional anemia) with mild increase in blood urea nitrogen. Was there any concern for Gastrointestinal bleeding?
In line 225 , the authors mentioned "CKD induced placental dysfunction" - the references provided were all for "pre-eclampsia" which the patient does not seem to have based on the presence of hypotension. Please provide references for CKD induced placental dysfunction if any in the absence of pre-eclampsia.
Want to share with the authors one study of moderate protein restricted plant based, supplemented diet in pregnant CKD patients which actually showed favorable improvement in utero-placental flows. Although the patients in the study only had CKD stage 2 and plant based diet was supplemented, unlike the patient in the current case report with advanced CKD and had a plant based diet that was not supplemented. https://academic.oup.com/ndt/article/33/suppl_1/i474/4998346
The case presentation is well written and the table and graphs are easily understandable. Need to work on discussion as above.
Author Response
Answer to reviewer 2:
Thank you for the suggestions to improve the quality of our paper; please find in the following lines (in red) the changes in the paper.
On page 8 , under "Plant based diets" line 171 - authors mention "small portion of animal-derived food 3-4 times per week at lunch time". What animal derived food was used? How much of the 2000Kcal/day calorie intake and 0.8g/ideal weight/day of protein intake per day did the animal derived food account for?
The following precisions were added, also om teh account of teh question of your co-reviewer:
This latter diet included small quantities of milk in the morning, and an occasional yogurt as a snack during the day, with muesli, cereal or biscuits, and a small portion of animal-derived food 3-4 times per week at lunchtime. Less than 20% of the protein intake is of animal origin, thus reducing the overall acid load. The fibber intake is high, but it was not specifically evaluated, due also to the indication to vary as much as possible the type of food, within the described indications.
Each main meal included cereal products (pasta, rice or other grains), beans, vegetables and fresh fruit, while dried fruit and nuts were eaten as snacks during the day.
In the absence of reference data on reference urinary urea in pregnancy, the protein intake was regularly assessed by the dietician, who controlled the patient at least monthly.
On page 9, line 225 and 226, the authors mention that the patient may have had "maternal pre-eclampsia" that led to the decision to induce delivery. Maternal Pre-eclampsia is a pregnancy complication characterized by high blood pressure and signs of damage to other organs such as liver and kidney. While the patient developed cholestasis and elevated creatinine, her blood pressure was only 90/55 mm hg suggesting against pre-eclampsia.
In line 225 , the authors mentioned "CKD induced placental dysfunction" - the references provided were all for "pre-eclampsia" which the patient does not seem to have based on the presence of hypotension. Please provide references for CKD induced placental dysfunction if any in the absence of pre-eclampsia.
While we agree with the reviewer that normotension is against a classic for of preeclampsia, we would like to stress that maternal PE is a form of PE associated with a pre-existing maternal disease and not necessarily with other maternal complications.
We tried to clarify it as follows:
The pattern is not classic superimposed PE, or “maternal preeclampsia”, a term often employed for defining the full blown picture of hypertension and proteinuria developing or worsening ion the basis of a pre-existing maternal disease (very often CKD). However, this pattern suggests a non-specific CKD-induced placental-dysfunction, quite common in CKD, where proteinuria often increases or develops in pregnancy, with or without hypertension, and is associated with an increased risk of preterm delivery and intrauterine growth restriction (6-7, 38-42). The presence of a worsening of the kidney function led to the decision to induce delivery, also on the account of search for a compromise between maternal and foetal health.
What other differential diagnoses were considered for evaluation of AKI? The patient's hemoglobin dropped by almost 2 units from 30 weeks to 33 weeks(although pregnancy can cause dilutional anemia) with mild increase in blood urea nitrogen. Was there any concern for Gastrointestinal bleeding?
No other cause of anemia was identified; you are right, and we added the following sentence: as well as worsening of anemia, in the absence of any sign of bleeding or inflammation.
Want to share with the authors one study of moderate protein restricted plant based, supplemented diet in pregnant CKD patients which actually showed favorable improvement in utero-placental flows. Although the patients in the study only had CKD stage 2 and plant based diet was supplemented, unlike the patient in the current case report with advanced CKD and had a plant based diet that was not supplemented. https://academic.oup.com/ndt/article/33/suppl_1/i474/4998346
Thank you so much for citing this small paper of our group; this is an issue on which we hope to work more in the future and we are presently updating the series. This is why we did not cite it, since it was for the moment just an abstract… Thanks again for your comments and suggestions, Sincerely the Authors.
Round 2
Reviewer 2 Report
All the comments to the authors are addressed in the revised manuscript
Author Response
thanks for your comments and suggestions.